# USING REINFORCEMENT LEARNING TO SOLVE VEHICLE ROUTING PROBLEMS WITH DYNAMIC CUSTOMERS

## ABSTRACT

Deep Machine Learning methods have been proven to be effective in solving the travelling salesman problem and general vehicle routing problems. In this paper, we use reinforcement learning to learn effective and fast constructive heuristics for solving a problem with dynamic customer requests, the partially dynamic travelling repairman problem, and variants with customer demands and time windows. We perform an ablation study on the policy network that maps between state and action spaces of arbitrary size to investigate which features from previous literature translate well to this new domain and introduce a recurrent neural network component to the decoder that tracks arrivals of dynamic customers improving performance on problems with time windows. The encoder-decoder network can map between state and action spaces of arbitrary dimension so we investigate how it generalizes to problems of different sizes. The performance of the construction heuristic is compared with several baselines on real world examples and different spatio-temporal customer request distributions of different sizes.

## 1 INTRODUCTION

We develop and deploy a graph neural network (GNN) encoder-decoder framework in a reinforcement learning approach to solving the partially dynamic travelling repairman problem (PDTRP; Larsen et al., 2002), an evolution of the travelling salesman problem in which customers arrive while the tour is being executed. Our work builds on Joshi et al. (2019), but introduces dynamism in the routing model to produce solutions that better capture the challenges faced in modern domains of application, such as repair call-outs, rapid courier services, restocking of industrial fuels, vehicle sharing services and emergency vehicle dispatch (Larsen et al., 2002; Pillac et al., 2013). We also extend the problem to include features, common in routing applications, such as time windows for service and customer demands, demonstrating the flexibility and efficacy of the learning framework.

Vehicle routing problems (VRPs) have been aiding dispatchers in planning routes in a wide range of logistical contexts since its inception as a generalization of the travelling salesman problem (Dantzig & Ramser, 1959). The PDTRP is an example of a dynamic vehicle routing problem (DVRP). DVRPs are distinguished from the more conventional VRPs by the input on the problem being received and updated concurrently with the determination of the route (Psaraftis et al., 2016). In practice these updates can have a dramatic effect on the success of a planned route. It is therefore useful to design solvers that, thanks to advances in communication technology, can make adaptations to a route in progress to react to dynamic information (Crainic et al., 2009).

A DVRP solver must be fast as solutions must be adapted in real-time in response to dynamic information. One of the most promising new methods for solving VRPs is to utilize reinforcement learning (RL) to learn policies that construct routes in a step-by-step fashion. This method has been shown to be competitive with exact solvers for the TSP and VRP (Kool et al., 2018). The iterative route construction lends itself very favourably to DVRPs because any dynamic information that arrives between steps can then be incorporated into the decision strategy at the next step. Furthermore, once trained, the policy can be queried very quickly which makes it ideal for use in real-time.

An interesting aspect of the PDTRP is that when modelled using a Markov Decision Process (MDP), the state space varies in size depending on the current number of customers awaiting service. When

applying the model to real situations, the total number of customers is unknown a priori. This means that the neural network architecture must be able to encode a state space of arbitrary size; we find a GNN to be an effective architecture for this purpose (Joshi et al., 2019). The action space is simply to select the next node to visit, and therefore also varies in size with the state space; our decoder takes an aggregate embedding of the full state space, and combines this with the embeddings of individual nodes through a transformer architecture to select the next node to visit. The problem requires the architecture to be agnostic to the state size, but can be trained efficiently on relatively small examples using masking and deployed on larger instances.

Reinforcement Learning based methods, and methods from the closely related field of approximate dynamic programming (ADP), have been applied to create solvers for a range of different DVRPs. These methods all begin by modelling the DVRP as an MDP. Ulmer et al. (2019) uses table lookup based value function approximation with online lookahead to solve a DVRP with stochastic customers where the aim is to maximize the number of customers served. ADP approaches that utilize linear function approximation and neural networks have also been utilized for DVRPs (Zhang et al., 2022; He et al., 2018). Zhou et al. (2023) combine supervised learning based on historical delivery routes with policy gradient methods. Iklassov et al. (2024a;b) apply reinforcement learning to DVRPs where the amount of customer demand and also travel times are dynamic. Akkerman et al. (2025) compare different ADP and RL approaches for DVRPs.

However a common theme across these DVRP papers is that they do not utilize many of the newest methods that apply sophisticated deep learning models to combinatorial optimization problems including routing. The catalyst for this stream of work is Vinyals et al. (2015) who tested their pointer network architecture on the TSP, training it to predict tours using supervised learning on optimal instances. Bello et al. (2016) trained the pointer network with policy gradient and actor critic RL to construct TSP tours in a step-by-step fashion and Nazari et al. (2018) applied a very similar approach to a single vehicle capacitated VRP. Both Deudon et al. (2018) and Kool et al. (2018) replaced the RNN based encoder in the pointer network policy with multi-headed attention (Vaswani et al., 2017) improving tour quality for the TSP. Kool et al. (2018) also developed a more performant critic baseline, a modified decoder, and demonstrated their method on the CVRP, VRP with split deliveries, the orienteering problem and the prize collecting TSP. In Joshi et al. (2019) the encoder is replaced with a graph neural network which gives another boost in performance.

Aside from making improvements to the model, there are several papers that propose new learning algorithms that improve performance. For constructive RL, Kwon et al. (2020), Kim et al. (2022) and Drakulic et al. (2023) improved performance by exploiting the symmetry present in RL solutions to combinatorial optimizations. Other deep learning paradigms other than RL have been explored for solving routing problems such as diffusion processes (Li et al., 2023) and unsupervised learning (Min et al., 2023). However, for the dynamic vehicle routing problems the RL based methods are the obvious choice due to their step-by-step solving aligning with the staggered arrival of new inputs.

In this paper, we develop an RL construction heuristic that can solve the PDTRP and variants with demands and time windows effectively, obtaining performance that is very close to or better than a sophisticated meta-heuristic based static solver (OR-Tools; Furnon & Perron, 2025) that is adapted to solve the PDTRP by solving a static "snapshot" of the problem every time the input is updated. Our method outperforms the heuristic methods suggested in the original paper (Larsen, 2000) and the method of Joshi et al. (2019) applied to static "snapshots" in the same manner as for OR-Tools. Notably, the RL construction heuristic produces its solutions very quickly with the time taken to solve PDTRP instances being comparable to the heuristic methods and orders of magnitude faster than the static resolving methods.

We also present an ablation study which tests the performance of different policy and learning components that have been developed in the literature for static problems and test some ideas we devised for improving performance in dynamic problems. We also demonstrate an ability to train on small instances and generalise to large problem instances that have not previously been experienced, which is particularly effective when there are no explicit time windows in the problem formulation.

## 2    PROBLEM DESCRIPTION

Adding dynamism to the already large taxonomy of VRPs has resulted in a similarly large taxonomy of DVRPs. Categories of DVRPs differ both in how dynamism presents itself and also how the solution can choose to react to dynamism. The main sources of dynamism in DVRPs are travel times, service times, demands and customer arrivals (Rios et al., 2021). There are problems in which the vehicle can choose to wait or reposition itself to better anticipate the arrival of dynamic information or reject dynamic requests that it will not be able to service given constraints upon it (Bent & Van Hentenryck, 2004). In others, including the PDTRP, the solver has the opportunity for limited anticipation through its choice of route but lacks these more powerful options.

Early solution methods for DVRPs were modified static solvers that either re-solved a static problem each time new input information appeared or modified a static solution in response to new information using an insertion heuristic (Psaraftis, 1988). This was made more efficient in Gendreau et al. (1999) where a tabu search meta-heuristic solves multiple routes in parallel to improve the quality of the solution obtained in the limited time between decision points. Bent & Van Hentenryck (2004) took this multiple plan approach and used probability models of future inputs or historical data (Bent & Van Hentenryck, 2005) to produce solutions that are better designed to receive dynamic customers. There are a number of other papers that apply meta-heuristics from VRPs to DVRPs by employing a resolving approach. For example, Schyns (2015) uses the ant colony optimization meta-heuristic, Sarasola et al. (2016) uses variable neighbourhood search and Novaes et al. (2015) uses genetic algorithms.

Some of the fastest solutions are achieved using heuristics such as the ones suggested in the original PDTRP paper (Larsen et al., 2002). The nearest neighbour and first come first served heuristics used here produce good quality solutions. Insertion heuristics have also been applied to DVRPs (Psaraftis, 1988), typically starting with the best static solution possible and letting the heuristic decide where to insert dynamic customers into the route.

In this article we focus on the PDTRP as defined by Larsen et al. (2002):

- A repairman travels at constant velocity, $u$, in a bounded region $\mathcal{A}$ of area $A$.

- A subset of customers are dynamic and arrive according to a Poisson process with parameter $\lambda$ between opening time at $t = 0$ and a time horizon $T$. The locations of the customers are independently and uniformly distributed in $\mathcal{A}$.

- An $N$ node instance consists of $N - 1$ customers plus a centrally located depot. $N_{adv}$ is the number of advanced request customers (plus the depot) present at the beginning of the problem. $N_{imm}$ is the number of immediate request customers that arrive dynamically. A key metric for the dynamism of a particular problem instance is the 'degree of dynamism (dod)', defined to be $N_{\text{imm}}/(N_{\text{adv}} + N_{\text{imm}})$.

- Each customer requires an independently and identically distributed amount of on-site service time $\delta_i$ that becomes known once the service is completed.

- The route is updated only at customer locations; a vehicle cannot change its destination while travelling.

- The objective is to minimize the repairman routing cost.

As we will apply reinforcement learning to this problem, we model it as an MDP. As is common practice for VRPs, the state, $s_k$, at each discrete time step, $k$, is represented as a graph, $\mathcal{G}_k = (V_k, E_k)$, where the nodes, $v_i \in V_k$, represent customers and are labelled with their co-ordinates in the unit square, $(x_i, y_i)$. The edges, $e_{ij} \in E_k$, represent paths that the repairman can traverse between customers and are labelled with the travel distance, $d_{ij}$, of the path. The action set, $A_k$, is the subset of nodes corresponding to customers who have not yet received service. The state also tracks the current time $t$. Each episode starts at a centrally located depot at time $t = 0$ and terminates when the repairman has serviced all customers, including dynamic ones, and returns to the depot.

Between time steps in the MDP, the repairman moves from node $v_i$ in state $s_k$ to node $v_{a_k}$ in state $s_{k+1}$, and $t$ is updated to $t' = d_{i,a_k}/u + \delta_{a_k}$. The state $s_k$ is updated to $s_{k+1}$ by adding any dynamic customers whose arrival time occurred in the interval $(t, t']$ to the graph. The objective function for

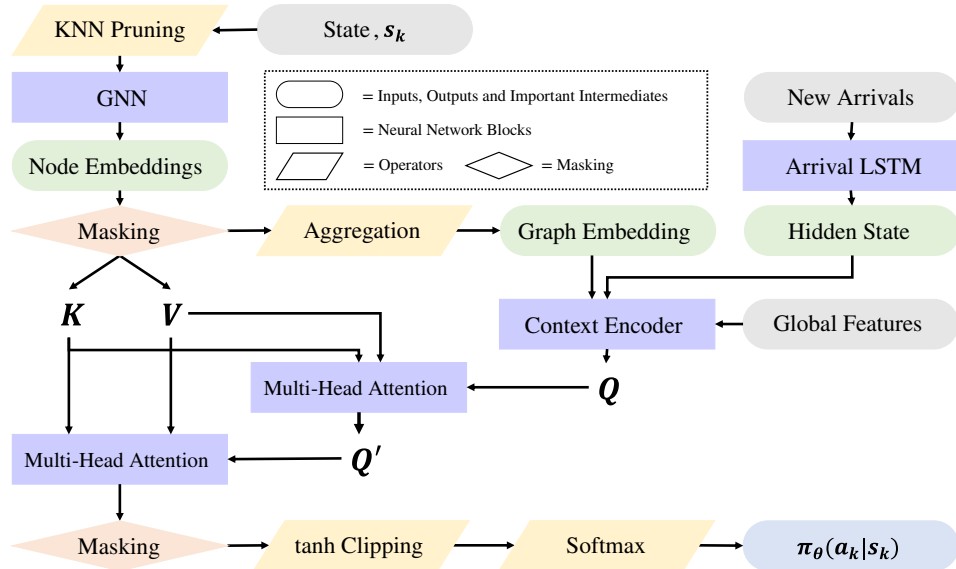

Figure 1: Illustration of the policy network architecture.

an instance of the PDTRP is defined as:

$$L = \sum_{i=0}^{N} d_{v_{\tau(i)}v_{\tau(i+1)}},$$

for a tour $\tau = (v_0, v_1, ..., v_0)$ consisting of $N+1$ total nodes where the starting and ending node are the depot $v_0$. For problems with time windows, we introduce an alternative objective function that penalizes lateness:

$$L = \sum_{i=0}^{N} d_{v_{\tau(i)}v_{\tau(i+1)}} + \sum_{i=1}^{N} \max(0, t_{\tau(i)} - l_{\tau(i)}),$$

when $t_{\tau(i)}$ is the time at which customer $i$ in the tour was visited and $l_{\tau(i)}$ is the end of that customer's time window.

For problems with customer demands, each customer has a randomly generated demand and in order to service a customer, the repairman must have a current capacity exceeding that demand. The repairman can return to the depot at any time to increase capacity. More details of these problems and their associated simulations are given in Appendix A.

## 3 REINFORCEMENT LEARNING

We use an actor-critic policy gradient algorithm (Sutton et al., 1999) to train a policy to construct solutions to these problems. The policy neural network is an Encoder-Decoder architecture and is depicted in Figure 1. In state, $s_k$, the graph $\mathcal{G}_k = (V_k, E_k)$ is input to the encoder which outputs a d-dimensional embedding of each node in the graph. The decoder component then takes this embedding and uses additional contextual inputs to output a probability distribution over active customer requests.

### 3.1 ENCODER

As in Joshi et al. (2020), the encoder consists of first pruning the input graph's edge set $E_t$ by using a k nearest neighbour algorithm (based on Joshi et al. (2019), k is equal to the $20\%$ of the total customers) to remove all edges that do not belong to a node's set of nearest neighbours.

After being pruned, the graph is passed to a GNN. In the GNN, the node and edge features are projected into $\mathbb{R}^d$ by linear layers to obtain feature vectors $\{h_i^0\}_{v_i \in V_k}$ and $\{e_{ij}^0\}_{e_{ij} \in E_k}$ which are

then passed through a stack of $L$ message passing layers. Each message passing layer updates each node's features by a weighted aggregation of its own features with those of its neighbouring nodes in the graph. The functional form of the message passing layers for node features is given by,

$$h_i^{\ell+1} = h_i^\ell + ReLU\left(\text{NORM}\left(U^\ell h_i^\ell + \text{AGGR}_{j \in \mathcal{N}_i}\left(\sigma(e_{ij}^\ell) \odot V^\ell h_j^\ell\right)\right)\right),$$

Here, $\ell$ indexes the message passing layer. $U^\ell$ and $V^\ell$ are network parameters, $\sigma$ indicates the application of a sigmoid function, AGGR indicates the application of an aggregation function over the set of neighbouring vertices $\mathcal{N}_i$ and NORM indicates the application of a batch normalization layer. The $\odot$ symbol is the element-wise multiplication operator.

The edge features are updated by a weighted aggregation of the edge features and the features of the two vertices connected by the edge. The functional form is given by,

$$e_{ij}^{\ell+1} = e_{ij}^\ell + ReLU\left(\text{NORM}(A^\ell e_{ij}^\ell + B^\ell h_i^\ell + C^\ell h_j^\ell\right),$$

$A^\ell, B^\ell$ and $C^\ell$ are network parameters. For a network with $L$ message passing layers, the final output of the encoder, $\{h_i^L\}_{v_i \in V_k}$, is a $d$-dimensional vertex embedding for each vertex in $\mathcal{G}_k$.

## 3.2 DECODER

The decoder component of the network builds from Kool et al. (2018). It operates sequentially, taking the encoder node embeddings at each time step $k$, and additional context, to produce a probability distribution over available customers.

From the encoder node embeddings, a graph embedding, $\bar{h}^L$, is obtained by aggregating over the node embeddings. This is concatenated with the embedding of the node that was visited by the repairman at the previous time step in the tour $h_{\tau(k-1)}^L$ to create a context vector, $h_c$. Depending on the problem being solved, we add more information to the decoder's context vector: for capacitated problems, the vehicle's current capacity; for problems with time windows, the current time $t$; we also experiment with adding the hidden state of a LSTM layer which takes as input the node features of any dynamic arrivals when they arrive in the problem.

The context vector $h_c$ is used as the sole query vector in a multi-headed attention (MHA) layer. The value and key vectors are the node embeddings $h_i^L$:

$$Q_c^m = W_Q^m h_c, \; K_i^m = W_K^m h_i^L, \; V_i^m = W_V^m h_i^L,$$

where $W_Q^m$, $W_K^m$ and $W_V^m$ are attention parameters for each attention head $m$. The output of this attention layer is an updated context vector $h_{c'}$,

$$h_{c'} = \sum_{m=1}^{M} W_m^O h_m,$$

$$h_m = \sum_i \frac{\exp\left(\frac{(Q_c^m)^T K_i^m}{\sqrt{d_K}}\right)}{\sum_j \exp\left(\frac{(Q_c^m)^T K_j^m}{\sqrt{d_K}}\right)} V_i^m,$$

where $W_m^O$ are parameters applied to the output of each of the M attention heads, which we index using $m$, and $d_K$ is the query/key dimension.

The updated context vector $h_{c'}$ is input to another MHA layer, again with M heads, but this time we are only interested in the attention weights. The attention weight of each node is taken and clipped to lie within the range $[-C, C]$ to obtain log-probabilities of each node:

$$u_i = C \cdot \tanh\left(\frac{Q_{c'}^T K_i}{\sqrt{d_k}}\right)$$

Here $Q_{c'} = W^Q h_{c'}$ is the query vector associated with the updated context and $K_i$ is the same key vector as in the previous layer. Nodes which are already part of the tour are masked; $u_i$ is set to $-\infty$ for these nodes. The log-probabilities are then input to a softmax function to obtain the probability of selecting each vertex:

$$\pi_\theta(a_k | s_k) = \frac{e^{u_i}}{\sum_j e^{u_j}}.$$

### 3.3 TRAINING

When training the model, we solve batches of simulated instances. Details of the simulations are given in Appendix A. The model is trained for one specific problem at a time as each problem has different node features and decoder contexts. Based on the results of the ablation study (Section 4.1), for problems without time windows, we utilize a "simplified" model that doesn't include features relating to dynamism like the arrival LSTM; this model has $\sim 350,000$ parameters. For problems with time windows, we utilize the full model with the arrival LSTM in the decoder context; this model has $\sim 500,000$ parameters.

The model is trained by a policy gradient algorithm with a unique critic baseline $b$, introduced by Kool et al. (2018), in which $b$ is obtained by running a cached policy in which actions are selected greedily rather than by sampling. The policy gradient is then given by:

$$\nabla L(\theta) = \frac{1}{M} \sum_{i=1}^{M} \sum_{(a_k, s_k) \in \tau} [(L(\tau) - b)\nabla \log \pi_\theta(a_k|s_k)].$$

At the end of an epoch, the cached policy is compared with the current policy and if the reward obtained by the current policy is a statistically significant improvement on the cached version, as determined by a $t$-test, then the cached model is replaced by the current policy. Fuller details of the training setup and parameters are given in Appendix C.

## 4 EXPERIMENTAL RESULTS

For the experimental results, we consider representative examples of the PDTRP problem:

**Uniform-uniform (U.U.) instances:** For this problem, we take the most straightforward approach to customer request generation and the one used in the original PDTRP paper. Customers' locations are generated uniformly in the unit square. Customer arrival times are generated uniformly between time 0 and the time horizon $T$.

**Skewed spatial-temporal (S.T.) instances:** For this example there is spatial and temporal dependency between the customer locations motivated by applications like courier services where we might expect customer requests to shift from commercial to residential areas over the course of a day. In these instances, the unit square is divided into four subregions, with different arrival rates in different subregions. Customers are more likely to be located in either the lower-left or upper-right subregion and dynamic customers in the lower-left are more likely to arrive in the first half of the time horizon, $[0, \frac{T}{2}]$ and customers in the upper-right are more likely to arrive in the second half of the time horizon, $[\frac{T}{2}, T]$. Exact details are provided in Appendix A.

**Real world (R.W.) instances:** These real world instances were provided in the EURO meets NeurIPS vehicle routing competition (Kool et al., 2022). The instances provided have an "explicit duration matrix providing (non-Euclidean) real world road driving times between customers". These instances are for the vehicle routing problem with time windows and therefore needed modifying to use them as test instances for the PDTRP and its variants. Details of the modification are given in Appendix A.2.

In each case we consider the standard PDTRP with no time windows or capacities, and in addition the same problem with time windows (PDTRPTW) and a capacitated version with time windows (PDCTRPTW). We also consider three degrees of dynamism (dod). Every model is trained on instances with dod ranging from $0.2 - 0.8$.

For the U.U. test instances, we present results for two learned models: a 20-50 model trained on U.U. instances with size between 20 and 50 nodes, and a 20-100 model trained on U.U. instances with size between 20 and 100 nodes. We use these two models and test on U.U. instances with $N = 50$ and $N = 100$ to test how well the model is able to generalize for each problem.

For the S.T. test instances, we present results for the 20-50 model trained on U.U. instances and a 20-50 model trained on S.T. instances. Similarly for the R.W. test instances, we present results for the 20-50 model trained on U.U. instances and a 20-50 model trained on R.W. instances. We choose

to test these models to see how performance differs between a "generic" model and one specifically trained for the test instances.

As comparison methods, we use the Nearest Neighbour (NN) and First Come First Served (FCFS) heuristics from Larsen et al. (2002). For each problem, we report results for the best performing of the two heuristics. We also use a resolving method based on OR Tools (Furnon & Perron, 2025). OR Tools contains a meta-heuristic based routing solver. Each time a customer request arrives, we resolve the problem with a 3 second time limit. Further details are in Appendix E. To compare our method with other RL approaches, we also apply a resolving approach with the method from Joshi et al. (2020) ('Joshi').

The table shows the average objective cost (obj.) over 100 test instances for each method applied to each problem. Recall the different objective costs for the problems with time windows which causes the objective costs to be higher for these problems. A percentage gap (% gap) is given to a static baseline where OR Tools is given 10 seconds to solve a static counterpart of each dynamic instance, in which all nodes are visible in advance (see Appendix F). The average time taken to solve an instance is reported. Hardware details and tables showing the results for PDCVRP and larger R.W. and S.T. instances are given in Appendix B.

The results for PDTRP on the U.U. test instances show that RL is able to quickly construct solutions that are only marginally worse than those produced by the OR Tools solver, in much faster times, and beats the other methods which produce solutions in a similar amount of time on most test sets. The performance of the RL policy compared to the OR tools baseline improves as the dod of instances becomes larger but for the other baselines it is more consistent suggesting this is due to the unsurprising fact that the method based on a static solver performs better on less dynamic instances.

For PDTRP, the models have good generalization performance with the $20 - 50$ model performing with almost parity to the $20 - 100$ model on instances with $N = 100$. The results also show that training on instances that are the same size as those seen in training confers an advantage and also that training on larger instances can cause performance to drop slightly on smaller instances, motivating a judicious choice of training instance sizes depending on the application.

The policy achieves remarkably good performance on PDTRPTW, even managing to beat the static benchmark. For the PDCTRPTW, the results achieve parity with OR Tools but do not match the performance we see for PDTRPTW. We hypothesise that this is a result of the capacity constraints restricting the choices of possible routes and leaving less room for the policy to adapt to the time windows.

When time windows are part of the problem, the model does not show the same generalization ability from small to large problems as in the PDTRP, although by increasing the training range of $N$ satisfactory results on a wider range of instance sizes can be achieved. We hypothesise that the weaker generalization on the problems with time windows is due to the scaling of instances in these problems. We keep the time horizon and vehicle speed the same whilst increasing the number of customers. This leads to a situation in which the vehicle will struggle to meet a lot of the time windows, hence the high objective costs across the board for the larger problems.

When looking at the R.W. instances, we see a similar pattern to the U.U. when it comes to comparing the performance of the model to the baselines, although in this case the model performance is noticeably better for the PDCTRPTW. We see that being trained on R.W. instances leads to a large performance boost over the model trained on U.U. instances.

### 4.1 ABLATION STUDY

We carried out an ablation study to determine which of the modelling innovations made in the literature for static VRPs are relevant for the dynamic setting and also which newly introduced model components for dealing with dynamic elements help model performance. When comparing training methods and edge features, we choose to use the simplest problem variation, the PDTRP. When we compared model features for the PDTRP, we didn't find much difference between the different models (see Appendix D). So we compare them for the PDCTRPTW instead.

For the training method comparison, we can clearly see that training on dynamic instances gave a noticeable boost in performance over training with static instances when we directly port the method

Table 1: Our method vs. baselines

| | Method | dod = 0.2 | | | dod = 0.5 | | | dod = 0.8 | | |
| | | Obj. | %Gap | Time | Obj. | %Gap | Time | Obj. | %Gap | Time |
|---|---|---|---|---|---|---|---|---|---|---|
| | | **Uniform - Uniform N = 50** | | | | | | | | |
| PDTRP | 20-50 | 9.30 | 63.63 | 0.07s | 14.06 | 147.4 | 0.10s | 18.51 | 225.7 | 0.11s |
| | 20-100 | 9.40 | 65.39 | 0.04s | 14.11 | 148.3 | 0.05s | 18.62 | 227.6 | 0.06s |
| | ORTools | **8.83** | 55.36 | 14.8s | **14.02** | 146.7 | 33.7s | **18.46** | 224.8 | 56.0s |
| | Joshi | 9.47 | 66.62 | 0.17s | 14.40 | 153.4 | 0.27s | 18.59 | 227.1 | 0.28s |
| | Heur. | 9.53 | 67.67 | 0.01s | 14.30 | 151.6 | 0.02s | 18.65 | 228.1 | 0.02s |
| PDTRP TW | 20-50 | **35.1** | -53.22 | 0.08s | **44.7** | -40.43 | 0.11s | 84.3 | 11.37 | 0.12s |
| | 20-100 | 46.0 | -38.69 | 0.04s | 47.1 | -37.23 | 0.05s | **75.9** | 0.28 | 0.06s |
| | ORTools | 69.8 | -6.97 | 25.5s | 92.6 | 23.41 | 47.8s | 121.6 | 60.65 | 60.0s |
| | Heur. | 11609 | 15372 | 0.04s | 10353 | 13698 | 0.04s | 9315 | 12206 | 0.04s |
| PDCTRP TW | 20-50 | 146.9 | 137.2 | 0.06s | 152.7 | 106.4 | 0.07s | 199.2 | 144.5 | 0.08s |
| | 20-100 | 98.1 | 58.4 | 0.06s | **100.2** | 35.4 | 0.07s | **119.3** | 46.4 | 0.08s |
| | ORTools | **58.2** | -6.03 | 27.6s | 110.4 | 49.2 | 57.4s | 137.4 | 68.7 | 74.8s |
| | Heur. | 12259 | 19694 | 0.08s | 11693 | 15703 | 0.10s | 9728 | 11334 | 0.07s |
| | | **Uniform - Uniform N = 100** | | | | | | | | |
| PDTRP | 20-50 | 12.56 | 59.12 | 0.15s | 15.21 | 92.67 | 0.19s | **17.72** | 124.5 | 0.21s |
| | 20-100 | 12.44 | 57.60 | 0.07s | 15.19 | 92.42 | 0.10s | 17.74 | 124.7 | 0.11s |
| | Joshi | 14.00 | 77.37 | 0.79s | 17.82 | 125.73 | 1.28s | 20.68 | 162.0 | 1.22s |
| | ORTools | **11.74** | 48.73 | 48.8s | **15.17** | 92.17 | 112s | 17.93 | 127.1 | 147s |
| | Heur. | 12.33 | 56.21 | 0.03s | **15.17** | 92.17 | 0.03s | 17.81 | 125.6 | 0.03s |
| PDTRP TW | 20-50 | 7413 | 359.2 | 0.17s | 7878 | 305.74 | 0.21s | 9021 | 268.2 | 0.23s |
| | 20-100 | 3891 | 141.0 | 0.08s | 4287 | 120.8 | 0.11s | 5057 | 106.4 | 0.12s |
| | ORTools | **1695** | 5.00 | 50.7s | **1988** | 2.39 | 97.0s | **2819** | 15.07 | 121s |
| | Heur. | 32819 | 1933 | 0.03s | 31213 | 1508 | 0.03s | 19251 | 686 | 0.03s |
| PDCTRP TW | 20-50 | 10754 | 280.7 | 0.11s | 10683 | 235.9 | 0.14s | 11294 | 212.9 | 0.15s |
| | 20-100 | 4929 | 74.47 | 0.11s | 5362 | 68.6 | 0.14s | 6175 | 71.1 | 0.15s |
| | ORTools | **4802** | 69.98 | 63s | **5233** | 64.5 | 102s | **5745** | 59.2 | 129s |
| | Heur. | 34280 | 1113 | 0.04s | 32653 | 926.6 | 0.08s | 30204 | 736.7 | 0.06s |
| | | **Real World Instances N = 50** | | | | | | | | |
| PDTRP | U.U. | 7.90 | 41.87 | 0.05s | 8.80 | 58.03 | 0.05s | 10.03 | 80.12 | 0.05s |
| | R.W. | 7.42 | 33.25 | 0.05s | 8.24 | 47.97 | 0.05s | 9.67 | 73.66 | 0.05s |
| | ORTools | **6.79** | 21.94 | 19.3s | 7.99 | 43.39 | 24.6s | 9.38 | 68.45 | 25.9s |
| | Heur. | 7.07 | 26.96 | 0.02s | **7.96** | 42.95 | 0.02s | **9.26** | 66.29 | 0.02s |
| PDTRP TW | U.U. | 2604 | -18.69 | 0.03s | 2579 | -19.14 | 0.04s | 3062 | -3.9 | 0.04s |
| | R.W. | **1377** | -57.0 | 0.03s | **1399** | -56.14 | 0.04s | **1994** | -37.42 | 0.04s |
| | ORTools | 3224 | 0.67 | 13.0s | 3201 | 0.36 | 15.8s | 3825 | 20.05 | 15.6s |
| | Heur. | 8883 | 177.4 | 0.01s | 8093 | 153.8 | 0.01s | 7867 | 146.9 | 0.01s |
| PDCTRP TW | U.U. | 4450 | 44.04 | 0.06s | 4269 | 36.35 | 0.06s | 4597 | 48.12 | 0.06s |
| | R.W. | **1867** | -39.57 | 0.05s | **1933** | -38.26 | 0.05s | **2405** | -22.51 | 0.05s |
| | ORTools | 3510 | 13.61 | 15.0s | 3783 | 20.83 | 18.7s | 4279 | 37.88 | 17.8s |
| | Heur. | 12259 | 296.8 | 0.03s | 11694 | 273.5 | 0.03s | 9728 | 213.5 | 0.03s |
| | | **Skewed Spatio-Temporal Instances N = 50** | | | | | | | | |
| PDTRP | U.U. | 8.69 | 63.13 | 0.09s | 12.26 | 130.1 | 0.12s | 15.87 | 197.8 | 0.14s |
| | S.T. | 8.44 | 58.43 | 0.04s | **12.03** | 125.8 | 0.05s | **15.76** | 195.8 | 0.06s |
| | ORTools | **8.19** | 53.74 | 14.5s | **12.03** | 125.8 | 31.1s | 15.93 | 199.0 | 48.5s |
| | Heur. | 8.71 | 63.5 | 0.02s | 12.30 | 130.8 | 0.02s | 15.91 | 198.6 | 0.02s |
| PDTRP TW | U.U. | **30.2** | -57.46 | 0.04s | **28.9** | -59.06 | 0.05s | **108.2** | 56.9 | 0.06s |
| | S.T. | 43.1 | -39.29 | 0.04s | 33.2 | -52.97 | 0.05s | 122.8 | 78.09 | 0.06s |
| | ORTools | 96.4 | 35.79 | 24.5s | 85.6 | 21.26 | 46.9s | 94.3 | 36.76 | 58.9s |
| | Heur. | 9075 | 12683 | 0.02s | 9220 | 12961 | 0.02s | 9735 | 14018 | 0.03s |
| PDCTRP TW | U.U. | 149.7 | 156.3 | 0.06s | 200.1 | 181.5 | 0.07s | 244.1 | 257.7 | 0.08s |
| | S.T. | **79.6** | 36.3 | 0.06s | **81.3** | 14.38 | 0.07s | 153.6 | 125.1 | 0.08s |
| | ORTools | 128.7 | 120.4 | 27.9s | 113.4 | 59.54 | 54.7s | **127.5** | 86.86 | 72.0s |
| | Heur. | 10196 | 17358 | 0.03s | 10352 | 14464 | 0.03s | 10461 | 15231 | 0.03s |

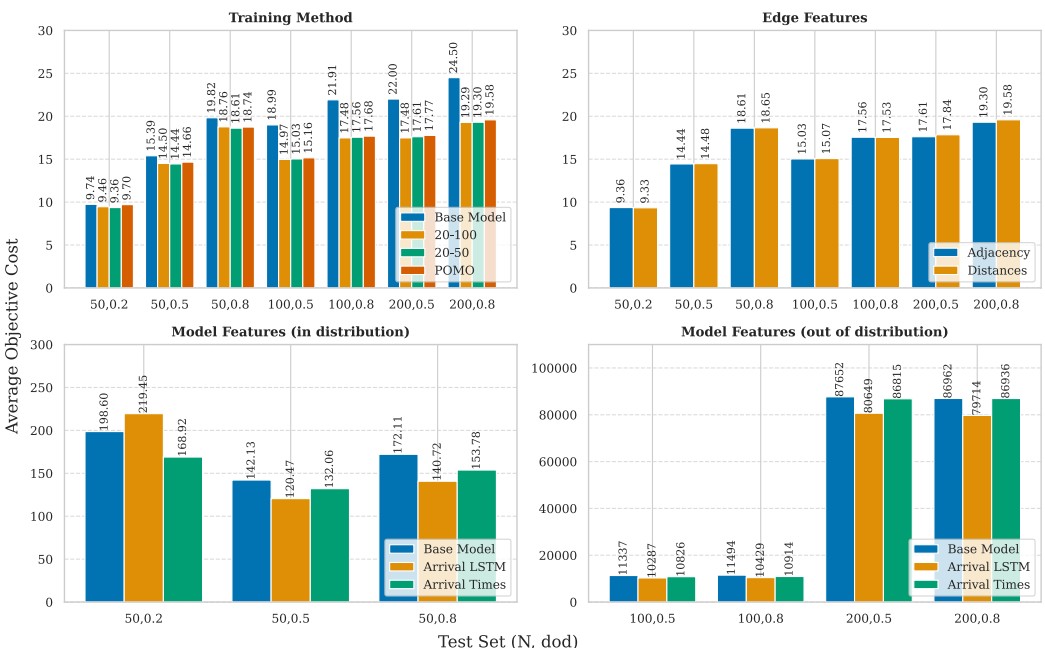

Figure 2: The ablation study was conducted on seven uniform-uniform test sets consisting of 64 instances each, each test is labelled with the $N$ and $DOD$ for its instances. The top-row's plots are instances of PDTRP and the bottom-row's are instances of PDCVRPTW.

from Joshi et al. (2019), which we consider to be the base model. Training on instances with a larger spread of $N$ gives better performance on larger datasets but slightly worse on the smaller ones. Implementing the POMO (Kwon et al., 2020) training method, in which single instances are trained on multiple times by choosing a different initial node (not counting the depot) each time and a different baseline is used, didn't give any noticeable performance boost although the training time on the same number of instances (counting each choice of starting node as an instance for POMO) compared to the other methods was shorter.

When looking at edge features, we compared an adjacency based edge feature which assigns every edge a fixed embedding vector with using distances, $d_{ij}$, as edge features. This likely indicates that the encoder çan learn to calculate the distances from the node locations. Since we find that the performance is effectively identical between these two embeddings, we suggest using the distance based embedding because for problems in which the distance between nodes does not correspond to a Euclidean distance, this edge feature will be the only way of supplying this information to the model.

Finally, we look at the results for the model features, this plot is split into in training distribution and out of training distribution performance due to the much larger costs associated with larger instances in this problem. Here we can see that including customer arrival times as part of the node features gives a boost to performance and this is further improved by adding a long short term memory RNN hidden state to the decoder context.

## 5 CONCLUSION

The results presented here demonstrate the effectiveness of deep reinforcement learning based constructive heuristics on the partially dynamic vehicle routing problem and its variants. Our key innovations are the use of reinforcement learning alongside GNN's to encode the node features in the network, aggregation to achieve summarise the network state, and use of a transformer to value each node in the context of the other nodes in the system. One of the great advantages of this framework is that it will remain broadly the same across similar applications with only changes needing to be made to how states and actions are represented to the policy. All code is freely available at XXXXX.

## 6 REPRODUCIBILITY STATEMENT

To aid reproducibility of our results, we have attached the code used to generate all of the results in this paper. The test datasets used to produce the results are available there and the training runs can even be reproduced by supplying the correct seeds. The appendix contains details of the simulations, training and baselines.

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

# A  PROBLEM SIMULATION

## A.1  UNIFORM-UNIFORM INSTANCES

We use the following procedure to generate uniform-uniform instances.

- Start with specified total number of nodes, $N$, or sample it uniformly at random between a specified minimum and maximum value. In training, once $N$ is set for a batch, it will apply to every instance in that batch.

- Generate the customer locations uniformly in the unit square by sampling the x and y coordinates from independent uniform distributions. The depot location is set at the center of the unit square.

- Generate arrival times for all customers by sampling uniformly on the interval $[0, T]$ where $T$ is the time horizon. For every training and test set in this paper, we have a time horizon of 8 hours.

- Generate customer service times by sampling from a log-normal distribution. For every training and test set in this paper, we set the parameters of the log-normal distribution to give a mean service time of 3 minutes with a standard deviation of 5 minutes.

- Get the number of immediate request customers by multiplying a specified dod by $N$ or sampling the dod uniformly between a specified minimum and maximum value and multiplying by $N$. The dod can vary between instances in a training batch.

- Get the number of advanced request customers $N_{adv} = N - N_{imm}$ and randomly sample $N_{adv}$ customers from the set of customers to be advanced request customers. The depot must be in the set of advanced requests.

- Set the arrival times of advanced request customers to 0

- For problems with time windows, add time windows following the procedure in Appendix A.4.

- For problems with demands, add demands following the procedure in Appendix A.5.

When generating instances and solving them, it is important that the vehicle velocity is specified. We use a vehicle velocity of 4 units per hour. Where 1 unit is the edge length of the unit square.

## A.2  REAL WORLD INSTANCES

The real world instances used in the paper are derived from a particular real world instance of the VRPTW from the Euro meets NeurIPS competition (Kool et al., 2022). This particular instance is included in the code repository accompanying this paper.

- Include the depot designated in the VRPTW instance as the first node. Sample $N - 1$ additional customers randomly from the customers in the VRPTW instance for a total of $N$ nodes.

- Divide the driving time matrix by the vehicle speed and then scale it so that locations lie within the unit square.

- Set the arrival time for customers equal to the start of their time window in the VRPTW and then scale the arrival times so that the locations which share the earliest arrival time now have arrival time 0 and are therefore treated as advanced request customers.

- scale the arrival times so that they all occur before the time horizon

- For problems with capacities, we generate demands in the same way as described in Appendix A.5.

- For problems with time windows, we take the time window starts and ends from the instance and shift them so that the starts coincide with their arrival times and the ends are scaled to be before the latest end time.

| IV | III |
|---|---|
| 1 | 5 |
| UNIFORM | LATE |
| I | II |
| 5 | 1 |
| EARLY | UNIFORM |

Figure 3: Diagram showing how the unit square is divided into subregions and assigned arrival weights and arrival skews for the skewed spatio-temporal test sets. Regions I and III have larger arrival weights and thus receive a larger share of customers that II and IV. The regions have arrival times generated by a *Beta* distribution. For regions labelled with *Uniform*, the parameters are ($\alpha = 1, \beta = 1$), for early, ($\alpha = 2, \beta = 5$), for late, ($\alpha = 5, \beta = 2$)

### A.3 SKEW SPATIO-TEMPORAL INSTANCES

Figure 3 shows the simulation scheme for customers in the skew spatio-temporal instances. In the code accompanying the paper, an unlimited number of different generation schemes can be created by varying the number of subregions, the weightings between subregions and the arrival time skews within subregions.

### A.4 TIME WINDOWS

For the PDTRPTW and PDCTRPTW, the time windows are generated based on a method from Larsen et al. (2004) as follows:

- A minimum and maximum time window duration is specified.

- A latest time window end is calculated by specifying an amount of time after the time horizon by which all time windows must be closed.

- For immediate customers the time window start is set to the customer arrival time plus a fixed reaction time. For advanced customers the time window start is sampled at any time between, the earliest possible service time, which is the travel time from the depot to the customer, and the latest time window end minus the minimum window duration.

- The time window ends are then generated by sampling the time window duration between the minimum and maximum for each customer and adding that to the time window start.

### A.5 VEHICLE CAPACITY AND DEMANDS

For the PDCTRP and PDCTRPTW, demands are generated as follows:

- The vehicle capacity is set to 1 for convenience.

- The minimum required number of trips that the repairman will need to make back to the depot to service every customer is sampled uniformly between a specified upper and lower bound.

- A total demand is sampled based on the minimum required number of trips.

- Raw demands are sampled for customers uniformly between a minimum demand and the capacity of the vehicle.

- The raw demands are then scaled so that their total matches the total demand.

Table 2: Our method vs. baselines for all uniform-uniform test sets

| | | Uniform - Uniform N = 50 | | | | | | | | |
| | | dod = 0.2 | | | dod = 0.5 | | | dod = 0.8 | | |
| | Method | Obj. | %Gap | Time | Obj. | %Gap | Time | Obj. | %Gap | Time |
|---|---|---|---|---|---|---|---|---|---|---|
| PDTRP | 20-50 | 9.30 | 63.63 | 0.07s | 14.06 | 147.4 | 0.10s | 18.51 | 225.7 | 0.11s |
| | 20-100 | 9.40 | 65.39 | 0.04s | 14.11 | 148.3 | 0.05s | 18.62 | 227.6 | 0.06s |
| | ORTools | **8.83** | 55.36 | 14.8s | **14.02** | 146.7 | 33.7s | **18.46** | 224.8 | 56.0s |
| | Joshi | 9.47 | 66.62 | 0.17s | 14.40 | 153.4 | 0.27s | 18.59 | 227.1 | 0.28s |
| | Heur. | 9.53 | 67.67 | 0.01s | 14.30 | 151.6 | 0.02s | 18.65 | 228.1 | 0.02s |
| PDCTRP | 20-50 | 12.24 | 46.43 | 0.11s | 15.86 | 89.43 | 0.14s | 19.40 | 131.6 | 0.16s |
| | 20-100 | 12.20 | 45.95 | 0.05s | 15.76 | 88.24 | 0.07s | 19.42 | 131.8 | 0.08s |
| | ORTools | **11.20** | 33.99 | 29.6s | **15.34** | 83.22 | 66.0s | **19.28** | 130.7 | 91.82s |
| | Heur. | 12.85 | 54.73 | 0.02s | 16.94 | 102.33 | 0.03s | 21.24 | 153.6 | 0.03s |
| PDTRP TW | 20-50 | **35.1** | -53.22 | 0.08s | **44.7** | -40.43 | 0.11s | 84.3 | 11.37 | 0.12s |
| | 20-100 | 46.0 | -38.69 | 0.04s | 47.1 | -37.23 | 0.05s | **75.9** | 0.28 | 0.06s |
| | ORTools | 69.8 | -6.97 | 25.5s | 92.6 | 23.41 | 47.8s | 121.6 | 60.65 | 60.0s |
| | Heur. | 11609 | 15372 | 0.04s | 10353 | 13698 | 0.04s | 9315 | 12206 | 0.04s |
| PDCTRP TW | 20-50 | 146.9 | 137.2 | 0.06s | 152.7 | 106.4 | 0.07s | 199.2 | 144.5 | 0.08s |
| | 20-100 | 98.1 | 58.4 | 0.06s | **100.2** | 35.4 | 0.07s | **119.3** | 46.4 | 0.08s |
| | ORTools | **58.2** | -6.03 | 27.6s | 110.4 | 49.2 | 57.4s | 137.4 | 68.7 | 74.8s |
| | Heur. | 12259 | 19694 | 0.08s | 11693 | 15703 | 0.10s | 9728 | 11334 | 0.07s |
| | | Uniform - Uniform N = 100 | | | | | | | | |
| PDTRP | 20-50 | 12.56 | 59.12 | 0.15s | 15.21 | 92.67 | 0.19s | **17.72** | 124.5 | 0.21s |
| | 20-100 | 12.44 | 57.60 | 0.07s | 15.19 | 92.42 | 0.10s | 17.74 | 124.7 | 0.11s |
| | Joshi | 14.00 | 77.37 | 0.79s | 17.82 | 125.73 | 1.28s | 20.68 | 162.0 | 1.22s |
| | ORTools | **11.74** | 48.73 | 48.8s | **15.17** | 92.17 | 112s | 17.93 | 127.1 | 147s |
| | Heur. | 12.33 | 56.21 | 0.03s | **15.17** | 92.17 | 0.03s | 17.81 | 125.6 | 0.03s |
| PDCTRP | 20-50 | 15.79 | 50.95 | 0.20s | 18.21 | 72.71 | 0.24s | 20.43 | 93.65 | 0.27s |
| | 20-100 | 15.36 | 46.84 | 0.10s | 17.59 | 66.83 | 0.12s | 19.88 | 88.44 | 0.14s |
| | ORTools | **13.98** | 33.64 | 53.9s | **16.70** | 58.39 | 113s | **19.23** | 82.28 | 150s |
| | Heur. | 15.45 | 47.70 | 0.04s | 17.72 | 68.06 | 0.05s | 19.94 | 89.01 | 0.04s |
| PDTRP TW | 20-50 | 7413 | 359.2 | 0.17s | 7878 | 305.74 | 0.21s | 9021 | 268.2 | 0.23s |
| | 20-100 | 3891 | 141.0 | 0.08s | 4287 | 120.8 | 0.11s | 5057 | 106.4 | 0.12s |
| | ORTools | **1695** | 5.00 | 50.7s | **1988** | 2.39 | 97.0s | **2819** | 15.07 | 121s |
| | Heur. | 32819 | 1933 | 0.03s | 31213 | 1508 | 0.03s | 19251 | 686 | 0.03s |
| PDCTRP TW | 20-50 | 10754 | 280.7 | 0.11s | 10683 | 235.9 | 0.14s | 11294 | 212.9 | 0.15s |
| | 20-100 | 4929 | 74.47 | 0.11s | 5362 | 68.6 | 0.14s | 6175 | 71.1 | 0.15s |
| | ORTools | **4802** | 69.98 | 63s | **5233** | 64.5 | 102s | **5745** | 59.2 | 129s |
| | Heur. | 34280 | 1113 | 0.04s | 32653 | 926.6 | 0.08s | 30204 | 736.7 | 0.06s |

## B    COMPLETE EXPERIMENTAL RESULTS

To obtain the timing results for the tables in this paper. The RL methods and Joshi baseline ran on a single *Nvidia L40 48G GPU*. Training of our models were also performed using this GPU. The OR Tools and heuristic baselines ran on a *Intel Xeon(R) Gold 6248R CPU @ 3.00GHz x 4*.

## C    TRAINING DETAILS

We train each model for 100 epochs consisting of 12, 800 instances per epoch. At the end of each epoch, the model solves 1280 instances greedily and it's average performance is compared with the performance of the cached baseline model using a t-test with $p = 0.05$. The dimension of the hidden layers in the encoder and decoder is set to 128. This affects the message passing and linear layers in the network. The normalization layers in the encoder are batch normalization layers with $\epsilon = 1e-5$, a momentum argument of 0.1 and learnable affine parameters. The number of heads in the MHA layers is 8. When time features are used, they are scaled by dividing by the time horizon. The Adam optimizer (Kingma & Ba, 2014) is used with a learning rate of $1e-5$. The grad norm is clipped to

Table 3: Our method vs. baselines for all real-world test sets

| | | Real World Instances N = 50 | | | | | | | | |
| | | dod = 0.2 | | | dod = 0.5 | | | dod = 0.8 | | |
| | Method | Obj. | %Gap | Time | Obj. | %Gap | Time | Obj. | %Gap | Time |
|---|---|---|---|---|---|---|---|---|---|---|
| PDTRP | U.U. | 7.90 | 41.87 | 0.05s | 8.80 | 58.03 | 0.05s | 10.03 | 80.12 | 0.05s |
| | R.W. | 7.42 | 33.25 | 0.05s | 8.24 | 47.97 | 0.05s | 9.67 | 73.66 | 0.05s |
| | ORTools | **6.79** | 21.94 | 19.3s | 7.99 | 43.39 | 24.6s | 9.38 | 68.45 | 25.9s |
| | Heur. | 7.07 | 26.96 | 0.02s | **7.96** | 42.95 | 0.02s | **9.26** | 66.29 | 0.02s |
| PDCTRP | U.U. | 11.58 | 37.46 | 0/07s | 12.16 | 45.18 | 0.07s | 12.93 | 55.20 | 0.07s |
| | R.W. | 10.59 | 25.70 | 0.05s | 11.02 | 31.57 | 0.05s | 11.83 | 41.90 | 0.05s |
| | ORTools | **9.39** | 11.46 | 19.7s | **10.13** | 20.95 | 24.8s | **11.27** | 35.27 | 25.9s |
| | Heur. | 10.61 | 25.94 | 0.02s | 11.42 | 36.35 | 0.02s | 12.34 | 48.11 | 0.03s |
| PDTRP TW | U.U. | 2604 | -18.69 | 0.03s | 2579 | -19.14 | 0.04s | 3062 | -3.90 | 0.04s |
| | R.W. | **1377** | -57.0 | 0.03s | **1399** | -56.14 | 0.04s | **1994** | -37.42 | 0.04s |
| | ORTools | 3224 | 0.67 | 13.0s | 3201 | 0.36 | 15.8s | 3825 | 20.05 | 15.6s |
| | Heur. | 8883 | 177.4 | 0.01s | 8093 | 153.8 | 0.01s | 7867 | 146.9 | 0.01s |
| PDCTRP TW | U.U. | 4450 | 44.04 | 0.06s | 4269 | 36.35 | 0.06s | 4597 | 48.12 | 0.06s |
| | R.W. | **1867** | -39.57 | 0.05s | **1933** | -38.26 | 0.05s | **2405** | -22.51 | 0.05s |
| | ORTools | 3510 | 13.61 | 15.0s | 3783 | 20.83 | 18.7s | 4279 | 37.88 | 17.8s |
| | Heur. | 12259 | 296.8 | 0.03s | 11694 | 273.5 | 0.03s | 9728 | 213.5 | 0.03s |
| | | Real World Instances N = 100 | | | | | | | | |
| PDTRP | U.U. | 12.33 | 54.97 | 0.09s | 12.92 | 62.49 | 0.09s | 13.68 | 72.00 | 0.09s |
| | R.W. | 11.27 | 41.65 | 0.09s | 11.71 | 47.27 | 0.09s | 12.53 | 57.55 | 0.09s |
| | ORTools | **8.78** | 10.35 | 23.6s | **9.68** | 21.74 | 26.6s | **10.46** | 31.52 | 27.1s |
| | Heur. | 9.68 | 21.67 | 0.02s | 10.27 | 29.16 | 0.02s | 11.11 | 36.69 | 0.02s |
| PDCTRP | U.U. | 16.13 | 54.89 | 0.12s | 16.71 | 60.66 | 0.13s | 17.38 | 66.69 | 0.13s |
| | R.W. | 14.55 | 39.72 | 0.08s | 14.97 | 43.93 | 0.08s | 15.61 | 49.72 | 0.09s |
| | ORTools | **11.13** | 6.88 | 23.6s | **11.77** | 13.16 | 26.8s | **12.60** | 20.85 | 27.1s |
| | Heur. | 12.81 | 23.01 | 0.04s | 13.44 | 29.22 | 0.04s | 14.32 | 37.34 | 0.04s |
| PDTRP TW | U.U. | 30739 | 15.17 | 0.06s | 30675 | 14.74 | 0.07s | 30545 | 13.54 | 0.07s |
| | R.W. | **26646** | -0.16 | 0.06s | **26699** | -0.07 | 0.07s | **27029** | 0.47 | 0.07s |
| | ORTools | 27363 | 2.52 | 15.6s | 27671 | 3.56 | 17.5s | 27918 | 3.78 | 18.0s |
| | Heur. | 32819 | 22.97 | 0.03s | 31213 | 16.82 | 0.03s | 29251 | 8.73 | 0.03s |
| PDCTRP TW | U.U. | 40753 | 49.49 | 0.11s | 39544 | 44.83 | 0.11s | 39147 | 43.43 | 0.11s |
| | R.W. | **28552** | 4.73 | 0.09s | **28612** | 4.79 | 0.09s | **28732** | 5.27 | 0.09s |
| | ORTools | 29573 | 8.48 | 19.8s | 29958 | 9.72 | 23.5s | 30711 | 12.52 | 20.8s |
| | Heur. | 34280 | 25.75 | 0.04s | 32653 | 19.60 | 0.04s | 30204 | 10.66 | 0.04s |

Table 4: Our method vs. baselines for all skew spatio-temporal test sets

| | Method | \multicolumn Skewed Spatio-temporal Instances N = 50 | | | | | | | | |
|---|---|---|---|---|---|---|---|---|---|---|
| | | dod = 0.2 | | | dod = 0.5 | | | dod = 0.8 | | |
| | | Obj. | %Gap | Time | Obj. | %Gap | Time | Obj. | %Gap | Time |
| PDTRP | U.U. | 8.69 | 63.13 | 0.09s | 12.26 | 130.1 | 0.12s | 15.87 | 197.8 | 0.14s |
| | S.T. | 8.44 | 58.43 | 0.04s | **12.03** | 125.8 | 0.05s | **15.76** | 195.8 | 0.06s |
| | ORTools | **8.19** | 53.74 | 14.5s | **12.03** | 125.8 | 31.1s | 15.93 | 199.0 | 48.5s |
| | Heur. | 8.71 | 63.5 | 0.02s | 12.30 | 130.8 | 0.02s | 15.91 | 198.6 | 0.02s |
| PDCTRP | U.U. | 11.69 | 42.11 | 0.10s | 14.51 | 77.33 | 0.12s | 17.89 | 117.6 | 0.14s |
| | S.T. | 11.40 | 38.58 | 0.05s | 14.17 | 73.18 | 0.07s | **17.52** | 113.1 | 0.08s |
| | ORTools | **10.62** | 29.10 | 29.5s | **14.06** | 71.83 | 66.5s | 17.66 | 114.8 | 93.8s |
| | Heur. | 12.24 | 48.79 | 0.02s | 15.48 | 89.19 | 0.03s | 19.06 | 131.8 | 0.03s |
| PDTRP TW | U.U. | **30.2** | -57.46 | 0.04s | **28.9** | -59.06 | 0.05s | **108.2** | 56.9 | 0.06s |
| | S.T. | 43.1 | -39.29 | 0.04s | 33.2 | -52.97 | 0.05s | 122.8 | 78.09 | 0.06s |
| | ORTools | 96.4 | 35.79 | 24.5s | 85.6 | 21.26 | 46.9s | 94.3 | 36.76 | 58.9s |
| | Heur. | 9075 | 12683 | 0.02s | 9220 | 12961 | 0.02s | 9735 | 14018 | 0.03s |
| PDCTRP TW | U.U. | 149.7 | 156.3 | 0.06s | 200.1 | 181.5 | 0.07s | 244.1 | 257.7 | 0.08s |
| | S.T. | **79.6** | 36.3 | 0.06s | **81.3** | 14.38 | 0.07s | 153.6 | 125.1 | 0.08s |
| | ORTools | 128.7 | 120.4 | 27.9s | 113.4 | 59.54 | 54.7s | **127.5** | 86.86 | 72.0s |
| | Heur. | 10196 | 17358 | 0.03s | 10352 | 14464 | 0.03s | 10461 | 15231 | 0.03s |
| | | \multicolumn Skewed Spatio-temporal Instances N = 100 | | | | | | | | |
| PDTRP | U.U. | 11.70 | 58.06 | 0.17s | 14.10 | 90.49 | 0.22s | 16.48 | 122.6 | 0.25s |
| | S.T. | 12.60 | 70.22 | 0.07s | 15.77 | 113.1 | 0.09s | 18.55 | 150.6 | 0.10s |
| | ORTools | **10.69** | 44.42 | 50.0s | **13.61** | **83.87** | 111s | 16.18 | 118.6 | 149s |
| | Heur. | 11.18 | 51.04 | 0.03s | 13.78 | 86.17 | 0.03s | **16.07** | 117.1 | 0.03s |
| PDCTRP | U.U. | 14.84 | 44.44 | 0.19s | 16.72 | 67.21 | 0.22s | 19.05 | 90.47 | 0.24s |
| | S.T. | 15.53 | 55.34 | 0.10s | 17.67 | 76.71 | 0.12s | 20.17 | 101.7 | 0.13s |
| | ORTools | **12.81** | 28.13 | 54.8s | **14.94** | 49.41 | 115s | **17.05** | 70.47 | 148s |
| | Heur. | 14.23 | 42.34 | 0.04s | 15.92 | 59.21 | 0.04s | 18.01 | 80.07 | 0.04s |
| PDTRP TW | U.U. | 6369 | 365.1 | 0.08s | 6920 | 302.1 | 0.11s | 7322 | 241.1 | 0.12s |
| | S.T. | 8344 | 509.3 | 0.08s | 8907 | 417.6 | 0.11s | 9426 | 339.1 | 0.11s |
| | ORTools | **1687** | 23.19 | 50.7s | **1864** | 8.32 | 95.7s | **2284** | 6.39 | 117s |
| | Heur. | 32239 | 2254 | 0.03s | 30492 | 1672 | 0.03s | 27248 | 1169 | 0.03s |
| PDCTRP TW | U.U. | 11153 | 336.5 | 0.12s | 11158 | 293.3 | 0.14s | 11922 | 270.4 | 0.15s |
| | S.T. | 10216 | 299.8 | 0.11s | 10211 | 260.0 | 0.13s | 11172 | 247.1 | 0.14s |
| | ORTools | **3750** | 46.76 | 55.5s | **4674** | 64.76 | 104s | **5391** | 67.49 | 126s |
| | Heur. | 34090 | 1234 | 0.04s | 32296 | 1038 | 0.05s | 28679 | 791.0 | 0.05s |

1.0. The function used for neighbourhood aggregation in the message passing layer was maximum and the function for aggregation to a graph embedding was mean.

During training, due to the requirement of most machine learning libraries to have dimensions fixed across a batch of training samples, the number of total customers, $N$, for instances in a batch is sampled uniformly between a minimum and maximum value and is fixed across instances in the batch. The number of dynamic customers, $N_{imm}$, is then sampled uniformly between a minimum and maximum degree of dynamism. The arrival times for the dynamic customers are then sampled from a Poisson distribution conditional on $N_{imm}$.

To stop the accumulation of gradient information during training (which can lead to out of memory errors), rather than passing the graph, $\mathcal{G}_k$, through the encoder at each time step, the graph containing nodes for every customer $v_i \in N$ is passed through the encoder at the start of the simulation of an instance. To obtain an appropriate encoding of the graph $\mathcal{G}_k$ at each time step, encodings of customers not present at that time step are masked before being passed to the decoder.

## D  ADDITIONAL ABLATION RESULTS

The results of the ablation test investigating the effect of model features in the PDTRP problem is shown in Figure 4.

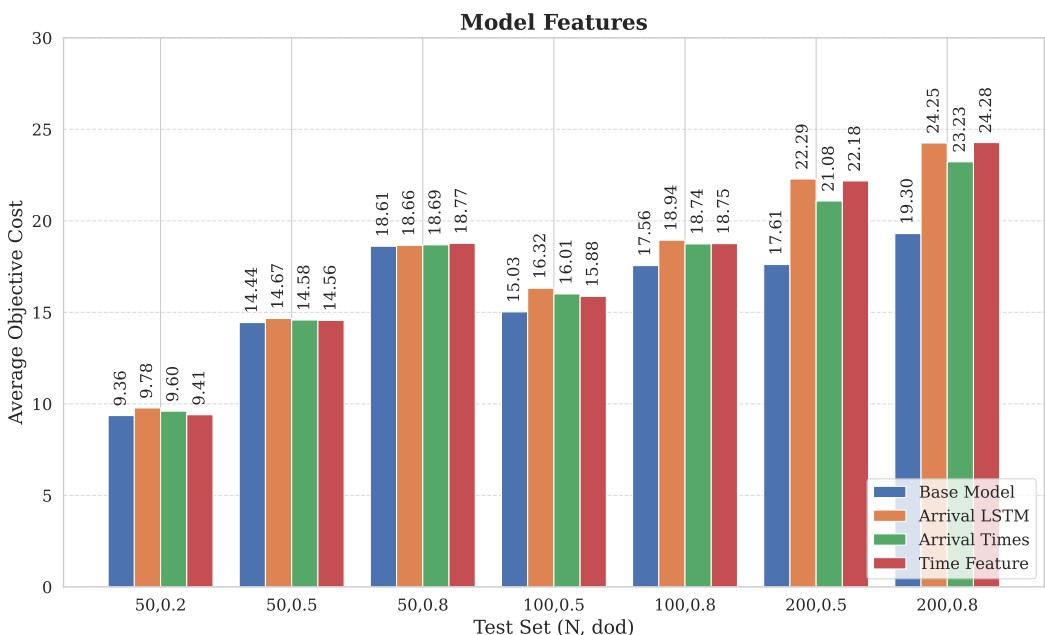

Figure 4: The ablation test for different model features on the PDTRP problem, notice that the base model performs best in all cases which is why we chose to utilize it for problems without time features in the experimental study. 'Time feature' included the current time in the decoder context for the PDTRP. The time feature was included by default for problems with time windows which is why this model feature doesn't appear for the PDCTRPTW ablation test.

## E  OR TOOLS RESOLVING

OR Tools (Furnon & Perron, 2025) is an open source optimization software suite that implements a wide range of meta-heuristic routing solvers such as guided local search, tabu search and simulated annealing and can be applied to most types of vehicle routing problem. It is possible to fix part of a route during the solving process which makes adapting it to dynamic problems an easier task. For the comparisons in this paper, for each re-solve, we use a first solution search which creates an

initial solution based on which edges are the most constrained and then spends 3 seconds improving that solution using guided local search.

# F   STATIC BASELINE

In the literature, a performance metric exists for online algorithms, called the competitive ratio (Sleator & Tarjan, 1985), that compares the cost of solutions found by an online algorithm with those found by an offline algorithm that had access to the entire instance, including dynamic requests. Following this line of thinking, we list the gap between each of our solution's average performance and the average performance of OR Tools applied to offline counterparts of each instance. The offline instances are generated by simply taking the online dynamic instances and setting the arrival times of all of the immediate request customers to 0 making them into advanced request customers. We then let OR Tools have 10 seconds to solve this offline instance.

Due to the complexity of VRPs, it is not possible to employ an exact solver even on the offline instances. This can lead to rare situations where this baseline is outperformed. We see a few cases of this with the PDTRPTW and PDCVRPTW where we get a negative % gap.

