# OpenReview forum: "Using reinforcement learning to solve vehicle routing problems with dynamic customers"
_ICLR.cc/2026/Conference — Submitted to ICLR 2026_

### Official Review · Reviewer_Q2AM · 2025-10-28

**Soundness:** 2
**Presentation:** 2
**Contribution:** 2
**Rating:** 4
**Confidence:** 4

**Summary:**

This paper presents a reinforcement learning framework for solving VRPs with dynamic customer requests. It considers the partially dynamic travelling repairman problem (PDTRP) and its variants with customer demands and time windows. The authors use a graph neural network (GNN) encoder-decoder architecture combined with a recurrent neural network in the decoder to handle dynamic customer arrivals and incorporate variable-sized state and action spaces. Their RL policy is trained to construct routes step-by-step and quickly adapt to new incoming requests in real-time, making the method suitable for dynamic, practical scenarios such as emergency dispatch, courier services, and industrial logistics.​

The paper includes an ablation study to test which network features and training methods improve performance, and demonstrates that their trained policies generalize from small to large problem instances. The framework is benchmarked against classical heuristics and state-of-the-art solvers (OR-Tools, previous RL and heuristic approaches), showing competitive and sometimes superior results on simulated and real-world dynamic datasets across multiple VRP variants.​

The contributions include: 1) Development of an RL-based constructive heuristic for dynamic VRPs, generalizing to arbitrary size and handling real-time dynamism. 2) Integration of GNNs and RNNs for state encoding and tracking dynamic arrivals, along with a transformer-based decoder for node selection. 3) Detailed ablation studies investigating model components and training strategies for dynamic routing. 4) Empirical demonstration of fast solution times and robust performance across uniform, skewed, and real-world instance distributions, making the method practical for time-sensitive routing applications.

**Strengths:**

The paper combines reinforcement learning with graph neural networks and recurrent neural architectures to solve vehicle routing problems with dynamic customer requests. It is a significant extension beyond prior work focused on static or semi-static VRPs. By tackling the partially dynamic travelling repairman problem and practical variants with time windows and customer demands, it broadens the scope of RL for online combinatorial optimization. The integration of a GNN encoder with RNN-based context tracking is a creative and technically meaningful development.​

The proposed method is benchmarked on both simulated and real-world datasets (including instances adapted from major routing competitions), compared against strong baselines like OR-Tools and recent RL/heuristic approaches, and supported by detailed ablation studies. The RL policy demonstrates competitive or superior performance, with solution times that are orders of magnitude faster than resolving static snapshots via conventional solvers. Generalization ability across problem sizes is clearly reported and validated.​

The paper is well written, with clear explanations of the problem setup and all architectural choices. Figures and tables report results and ablation findings, facilitating understanding of why certain modeling decisions matter. Key challenges of dynamic and uncertain routing are contextualized with practical motivation throughout.​

The significance is high. The proposed RL method brings strong real-time optimization capability to dynamic VRPs, potentially improving routing in domains such as courier logistics, vehicle dispatch, and time-sensitive delivery. The robust generalization and adaptability make it valuable for practice, not just theory.

**Weaknesses:**

1. Although the RL policy generalizes from small to large problem instances within the tested range, there is limited evaluation on very large-scale problems (e.g., thousands of dynamic customers), heterogeneous fleets, or highly complex real-world scenarios with richer, evolving features. Performance is not assessed outside of stylized or competition-like settings, which raises questions about robustness in diverse and practical deployments.​

2. The comparison set relies mainly on classical heuristics, and OR-Tools with snapshot re-solving. More advanced or recent RL, ADP, or local search methods are not comprehensively benchmarked, especially for the most demanding dynamic and constrained scenarios. Thus, conclusions about superior or state-of-the-art performance may not be fully substantiated for the broader DVRP literature.​

3. Results show that the model’s performance can quickly degrade when tested on problem distributions or instance sizes that are mismatched with the training set. This suggests a lack of universal adaptability, and may imply a need for frequent retraining or fine-tuning as operational conditions or customer demand profiles change.​

4. The design and behavior of the RL policy, especially why specific architectural choices (like the transformer decoder components) yield benefits, and how routing decisions evolve over time, are not deeply analyzed or visualized. Real-world users may find the policy’s underlying logic opaque, making trust and debugging difficult.​

5. While the method is fast in simulation, there is no demonstration of robust integration or deployment in real-time vehicle dispatch or distributed logistics systems, where networking, synchronization, and actuator delays can present new failure modes not captured in the static or batched testing protocol.​

6. On the most constrained or time-sensitive problem variants (e.g., with tight time windows and capacities), the RL approach sometimes underperforms or is only on par with adapted classical solvers. This weakness is acknowledged but not deeply investigated; further ablations and failure case exploration would clarify root causes and improvement strategies.

**Questions:**

1. How does the policy handle catastrophic distribution shifts or customer arrival processes significantly different from training time, and what theoretical guarantees can you offer for real-world robustness?​

2. Can the authors explain the learning dynamics or provide ablation evidence on how the recurrent decoder and GNN encoder specifically contribute to generalization for much larger or more dynamic instances?​

3. What are the core scaling bottlenecks (computation, memory, or policy degradation) when moving to real-time, distributed fleet management with thousands of dynamic requests?

---

### Official Review · Reviewer_gjnc · 2025-10-31

**Soundness:** 2
**Presentation:** 2
**Contribution:** 1
**Rating:** 2
**Confidence:** 4

**Summary:**

The authors apply reinforcement learning to the "partially dynamic travelling repairman problem" (PDTRP). They use a graph neural network (GNNs) encoder-decoder framework to model state spaces of arbitrary size and include the hidden state of a Long-Short-Term-Memory (LSTM) layer, adding its output to the decoder's context vector.
For evaluation, they run OR Tools on static variants of the instances (i.e., with perfect information) as a benchmark, and include additional heuristic baselines for the dynamic instances.

**Strengths:**

- The routing problem tackled is an interesting one, as it includes uncertainty and is closer to real-world problems than most of the problems studied in the neural combinatorial optimization (NCO) literature.
- The arguing for using GNNs to model state spaces of arbitrary size is nice, again promising flexibility that could be valuable in routing problems closer to reality.

**Weaknesses:**

- Results
	- It is hard to objectively judge the method's performance, considering that the benchmark mentioned in the text (OR Tools on a "static counterpart of each dynamic instance", run for 10s) is (1) not actually included in the result table and (2) only run for 10 seconds, raising doubts about the suitability of the values obtained as a benchmark. Gaps of -59% are reported in the result table, raising the questions whether maybe better baselines could have been found, considering this "static counterpart" has perfect information and zero uncertainty.
	- Results for the Skewed spatial-temporal (S.T.) instances are not mentioned at all in the text.
- Structure
	- Some passages of the introduction are essentially listings of previous works. By restructuring the introduction, the motivation for this work could be presented to the reader with greater clarity.
	- The authors could state more clearly, e.g., at the end of the introduction as bullet points, what their main contributions in the paper are.
	- The structure of the paper would have benefited of a "related works" section, ideally with one subsection on the RL part and one on DVRPs (incl. the PDTRP). As it is, related works are scattered across the introduction and problem description sections, making for a somewhat chaotic read.
	- Concepts like LSTM and multi-head attention should briefly be introduced if they are to be used.
- Problem definition
	- For someone not familiar with the PDTRP, the problem description in section 2 should be more detailed. For example, it is never explained what the time horizon $T$ actually is, and when speaking of arrival times, often the time of arrival at a customer is meant, but here it seems to be the time at which a new customer is added to the system.
	- It could also be stated more clearly already in the problem definition on p. 3 that while customer nodes have service times, they do not per default have time windows (first explicitly mentioned at the bottom of p. 6).
- Form
	- Tables and Figures need to be referenced in the text!
	- From the main text, the reader has no idea how training was performed. Not even how many instances were used (except for the 100 test instances per problem configuration that served for the result table).
	- Figure 1 could be improved: Considering it's supposed to show the encoder-decoder architecture, it would be helpful to the reader if it would be marked which parts actually belong to the encoder, etc. It could also help the understanding of section 3.
	- Math formulas could have used more care:
		- p. 3: In the definition of $t'$ there are two issues, (1) $t$ i not included, but the current time should always be updated based on the last current time, and (2) $u$ is never introduced and it is not clear what $u$ is (probably for scaling the distances to the unit square?)
		- p. 4: Index $i$ in the calculation of $L$ should have run from $0$ to $N-1$ (there are $N-1$ customer nodes, starting the count at 1, not 0). Alternatively, $i$ could run from $1$ to $N$, but as it is, $v_{\tau(i+1)}$ would result in an invalid index.
		- p. 5: $T$ is not introduced in section 3, I assume it is not the time horizon
		- p. 6: $\tau$ was introduced as a concatenation of nodes, it has no direct information on actions and states
	- From reading section 3, there are several things that are unclear in the formulas, where the authors should have provided more extensive explanations (e.g., regarding network parameters for the GNNs). If explanations were kept overly short due to space limitations, maybe some parts should have been put in the appendix instead.

**Questions:**

- In the conclusion you write your "key innovations are the use of reinforcement learning alongside GNN's to encode the node features in the network, aggregation to achieve summarise in the network state, and use of a transformer to value each node in the context of the other nodes in the system". Considering that there have been papers combining GNNs and RLs at least since 2023, how are your contributions new or differ from existing neural combinatorial optimization research?
- What was the basic experiment setup? E.g., instances, hardware, learning rate, etc.
- p. 6, l. 318-321: How does this qualify as generalizing if problems of the same size were already seen during training (e.g., instances with 50 nodes in the 20-50 case)?
- p. 6, l. 322 ff.: What about the model trained on 20-100, why not include these as well for the S.T. and R.W. test instances, like they were for the U.U. test instances?
- Why are the benchmark values for OR Toolson the "static counterpart" not reported in the table?
- Why were Nearest Neighbour (NN) and First Come First Served (FCFS) heuristics combined into one line "Heur." in the table instead of simply reporting them in two separate rows in the table?
- p. 7. l. 343 ff.: This is not about the static OR Tools benchmark, is it? Make it clearer for the reader.
- How do you explain that 20-100 consistently performs better for PDCTRPTW on the U.U. N=50 case even though 20-50 could be considered more "in-distribution"?
- How do you explain that the gaps for PDTRPTW and PDCTRPTW behave inconsistently for S.T. N=50 instances, i.e., gaps going down for dod 0.5 (compared to dod 0.2), but then going drastically up again for dod 0.8?
- Figure 2, Training Method (top left, PDTRP): Why does (100, 0.8) have lower gaps on average (for 20-100, 20-50, and POMO) than (50, 0.8), even though the dod is the same between these two scenarios? Intuitively, one would expect longer routes for larger problems.

---

### Official Review · Reviewer_rdoa · 2025-10-31

**Soundness:** 2
**Presentation:** 3
**Contribution:** 2
**Rating:** 2
**Confidence:** 4

**Summary:**

The paper addresses the dynamic traveling salesman problem (TSP) where customers
appear during route execution, such that the route has to be revised dynamically.
The authors propose to add the new nodes to the state graph and then use the
graph convolutional network (GCN) based model from Joshi et al. 2019 to predict the
next customer to visit, but to train this model via reinforcement learning. They
also add an additional encoder component for the sequence of customers arrived
so far. In experiments on different variations of the problem (time windows, capacities)
they show that their model outperforms heuristics from the literature, rerouting
via OR tools and the vanilla GCN model.

**Strengths:**

- s1. The paper addresses an interesting, (compared to the static variants) not so frequently
  researched combinatorial optimization problem (COP): dynamic TSPs.
- s2. The paper is clearly written.

**Weaknesses:**

- w1. it is not clear why the heuristics perform so badly for variations with time windows.
- w2. it is not clear if the OR solvers have been seeded with a repaired instance, or have to
  solve the remaining problem from scratch.
- w3. a local search based neural solver starting from a heuristically repaired solution
  is a trivial baseline, but not compared.
- w4. better baseline OR solvers have been missed.
- w5. the methodological contribution is limited, the exact delineation to existing models not
  very explicit.
- w6. it is not clear why the ablation study for the RNN encoder of past arrivals shows an
  effect in the uniform scenario.

more details:
w1. it is not clear why the heuristics perform so badly for variations with time windows.
- Can you clarify: does this happen because both, nearest neighbor and first come first served
  do not respect time windows and thus incur very high penalties?
- Is the most basic insertion heuristic, to check all insertion points and choose the
  one with lowest cost, evaluated?

w2. it is not clear if the OR solvers have been seeded with a repaired instance, or have to
solve the remaining problem from scratch.
- Starting from a repaired solution and then just refining this a little bit seems so much
  less effort than construcing them from scratch again.

w3. a local search based neural solver starting from a heuristically repaired solution
is a trivial baseline, but not compared.
- same rational as w2. And there are many local-based neural solvers available
  who could address the problem without modification.

w4. better baseline OR solvers have been missed.
- Why do you compare against OR tools?
- LKH and HGS in most papers are found to provide way better results.

w5. the methodological contribution is limited, the exact delineation to existing models not
very explicit.
- Combining GCN encoders with RL is very close to POMO. You just say that the specific
  idea of POMO, to start from different points, did not help.

w6. it is not clear why the ablation study for the RNN encoder of past arrivals shows an
effect in the uniform scenario.
- In the ablation study in fig. 2 you look at the problem where new customers are
  distributed uniformly over time. To me it is not clear, what can the sequence of
  past customer arrivals tell the model in this case?
- I would understand the effect for the time-variant arrival distributions (called
  "skewed spatial-temporal" in your paper).

**Questions:**

- q1. Why do the repair heuristics perform so badly for problems with time windows?
- q2. Do OR solvers refine a repaired solution or try to solve problems from scratch?
- q3. Can you compare against a local search based neural solver that tries to
  improve the heuristically repaired solution?

---

### Official Review · Reviewer_WgWg · 2025-10-31

**Soundness:** 2
**Presentation:** 2
**Contribution:** 1
**Rating:** 2
**Confidence:** 5

**Summary:**

This paper applies reinforcement learning (RL) to the partially dynamic traveling repairman problem (PDTRP) and related variants with customer demands and time windows. The authors propose an encoder–decoder network that maps variable-sized state and action spaces, with a recurrent decoder component that tracks dynamic customer arrivals. The goal is to learn fast constructive heuristics that generalize across instance sizes. Empirical evaluation compares the learned heuristic to baselines, including OR-Tools in a rolling-horizon setting, on several real-world–inspired datasets.

**Strengths:**

1. The topic—learning constructive heuristics for dynamic routing—is relevant to both operations research and reinforcement learning communities.

2. The problem choice (partially dynamic traveling repairman) has practical interest, and the integration of recurrent decoding for dynamic requests is conceptually reasonable.

3. The paper attempts to conduct ablation studies to understand design choices, which shows some experimental care.

**Weaknesses:**

1. Lack of methodological novelty: The proposed encoder–decoder architecture closely follows prior works on neural combinatorial optimization and RL-based routing. No clear algorithmic or architectural innovation is introduced. The RNN-based decoder extension is minor and does not constitute a substantial methodological advance.

2. Weak experimental design and baselines: The choice and configuration of baselines are inappropriate for the studied problem.
• OR-Tools is a general-purpose solver not tailored to dynamic routing, and its use in a rolling-horizon fashion is a weak benchmark.
• Restricting OR-Tools to 3-second or 10-second runtime windows heavily biases results in favor of the proposed approach.
• A much stronger baseline would be a stochastic or sample-average approximation (SAA) method or other optimization-based dynamic repairman heuristics, which can easily be computed for the proposed problem.
Consequently, the empirical conclusions are unreliable.

3. Unclear problem framing: The relationship between the studied variant (PDTRP) and prior definitions in the literature is not well explained. It is unclear which benchmark instances are used or why those from previous foundational works are omitted. Without a well-defined benchmark setting, comparisons lose validity.

4. Inconsistent and incomplete reporting: The paper does not define some abbreviations or baseline methods (e.g., “Heur.” in Table 1). Figures and tables lack clear descriptions, and runtime constraints for different methods are inconsistently justified.

5. Limited generalization and insight: The paper neither provides theoretical insights nor demonstrates meaningful generalization. It remains a straightforward application of existing neural routing architectures to a modified problem setting.

6. Overall contribution below ICLR standards: Both in terms of technical innovation and experimental rigor, the work falls short of what is expected for a top-tier ML conference. The study could be better suited for a specialized application venue if reformulated and empirically strengthened.

**Questions:**

1. What concrete architectural innovations distinguish your encoder–decoder from prior works?

2. Why was OR-Tools chosen as the main baseline, and why were time limits set so restrictively?

3. Which specific benchmark instances from the literature were used or modified, and why are original PDTRP instances not employed?

4. Could the authors clarify what “Heur.” in Table 1 refers to?

5. How sensitive are the results to the choice of rolling-horizon interval and computational cutoff?

---

### Meta-Review · Area_Chair_EnJY · 2025-12-12

**Summary:**

All the four reviewers hold a negative rating (4,2,2,2). They pointed out that the proposed approach looks like a straightforward application of existing GNN/RL based encoder–decoder frameworks to a dynamic VRP variant, with only minor architectural extensions. Moreover, they thought the used baselines are weak and some settings are unfair. I agree with those raised concerns, and believe this paper cannot be accepted, especially given that the authors did not rebuttal.

**Reviewer Concerns:**

The authors did not rebuttal, so I believe all the above concerns are still outstanding.

**Reviewer Scores:**

The reviewers will not change the score (4,2,2,2) since the authors did not rebuttal.

---

### Decision · Program_Chairs · 2026-01-26

Reject